# Associations Between Screen Time, Physical Activity, and Sleep Patterns in Children Aged 3–7 Years—A Multicentric Cohort Study in Urban Environment

**DOI:** 10.3390/sports13040091

**Published:** 2025-03-21

**Authors:** Paula Torres, Ana Pablos, Laura Elvira, Diego Ceca, Michael Chia, Florentino Huertas

**Affiliations:** 1Doctoral School, Catholic University of Valencia San Vicente Mártir, 46001 Valencia, Spain; paula.torres@mail.ucv.es; 2Faculty of Physical Activity and Sport Sciences, Catholic University of Valencia San Vicente Mártir, 46900 Torrent, Spain; laura.elvira@ucv.es (L.E.); florentino.huertas@ucv.es (F.H.); 3Faculty of Educational Sciences, Valencian International University—VIU, 46002 Valencia, Spain; diego.ceca@professor.universidadviu.com; 4Physical Education and Sport Sciences, National Institute of Education, Nanyang Technological University, Singapore 637616, Singapore; michael.chia@nie.edu.sg

**Keywords:** screen time, physical exercise, sleep time, children, healthy habits

## Abstract

In most developed countries, children’s use of digital media has increased significantly. Concerns about how screen time (ST) affects physical activity (PA), sleep patterns (SL), and overall health habits have prompted further exploration of these associations. This study examined ST, PA, and SL patterns in children aged 3–7 years living in an urban environment. A multicenter prospective cohort study was conducted using the Surveillance of Digital Media Habits in Early Childhood Questionnaire (SMALLQ^®^). Parents of 243 children completed the questionnaire, providing data on their children’s ST, SL, and PA habits during weekdays and weekends. A series of ANOVA tests were performed to assess differences in weekday and weekend ST, PA, and SL across age groups and sexes, as well as to compare these results with UN recommendations. The findings revealed a non-significant trend in ST and a significant effect of age group on PA during both weekdays and weekends, as well as on weekend SL. No significant differences based on sex were observed. Additionally, the comparison indicated that 3–5-year-old preschool children exceeded the recommended ST during weekends, while 6–7-year-old first-grade elementary children failed to meet the PA recommendations. These results highlight the critical role of age-related changes in shaping PA and SL behaviors in young children, emphasizing the importance of targeted interventions to foster healthy habits in early childhood.

## 1. Introduction

In most developed countries, children’s exposure to digital media is a pervasive phenomenon, both within and outside the school environment [1]. The term “digital media exposure” refers to the consumption of digital content through visual, verbal, or multimodal devices [2], with the latter often categorized as “screen media”. Screen media exposure encompasses the time spent engaging with any type of screen, whether interactive (e.g., smartphones, tablets, video games, computers, or other portable technologies) or non-interactive (e.g., television) [3].

Numerous studies have documented the negative physical and psychological consequences of excessive screen use. Prolonged exposure to digital screens has been associated with sedentary behaviors, which are strongly linked to reduced physical fitness, increased obesity, elevated blood pressure, and a higher risk of metabolic syndrome in children and adolescents [4,5]. In fact, it has been reported that nearly 40 million children under the age of five are affected by childhood obesity [6]. This excessive screen use can also lead to disrupted sleep patterns and reduced physical activity (PA), forming a cycle of diminished PA, poor sleep, and excessive screen use [7,8,9]. Furthermore, excessive screen use in children aged 0 to 7 years has been correlated with aggressive behavior and poorer outcomes in reading comprehension, short-term memory, language acquisition, and vocabulary development [8]. Moreover, excessive ST during a child’s first year has been associated with an increased risk of developmental delays by age two, particularly in communication skills, and similar impacts have been observed among teenagers and young adults [10,11]. These developmental impairments can contribute to disordered eating patterns, academic difficulties [12], and diminished executive functioning [13], which can compromise educational outcomes in both the medium and long term.

Some authors emphasized that one of the reasons PA levels tend to decrease as children grow older is due to the academic demands of school, which often limit opportunities for physical play [14]. This age range is a period where children move from more unstructured play into a more rigid academic schedule, reducing the time available for PA [15]. Others similarly highlight the importance of unstructured play in promoting PA in young children [16]. Their work suggests that engaging in free, unstructured activities is vital for fostering a healthy activity level in early childhood. This shift may also influence children’s screen time (ST) and sleep routines, making the preschool-to-elementary transition a critical period for intervention.

A deeper understanding of this cycle and how it specifically impacts children aged 3–7 years is needed, particularly since some studies have linked excessive ST with negative outcomes like sleep deprivation, which is particularly concerning at this developmental stage [17,18]. Most studies linking digital media use with sedentary lifestyles in preschoolers emphasize the risks of excessive usage without categorically condemning it. Among the contributing factors, excessive digital media use combined with reduced time allocated to PA or sports is particularly significant.

Despite these risks, some research highlights the potential benefits of technology use among toddlers, particularly when employed for educational purposes through tools such as computers, tablets, and smartphones [2]. This research has demonstrated advantages in social and intellectual well-being, such as through online learning, and its utility in improving communication skill development [19], as well as facilitating communication for children with disabilities [20]. Given these conflicting findings, further studies are needed to elucidate the effects of digital media and address whether the central issue lies in excessive ST, the nature of screen usage, or other contributing variables.

Due to this divergence in perspectives and the existing evidence with unanswered questions, some researchers have sought to determine how ST truly impacts sleep time (SL) and PA. Many studies on childhood behavior focus on either younger children (under 3 years) or older children (8 years and above) [4,12,13]. However, limited research examines children aged 3–7, a crucial period for establishing lifestyle habits related to ST, PA, and SL, because it represents a time of rapid cognitive and physical development [15]. During this time, routines and lifestyles are formed, providing the foundation for future habits, even though they may evolve over time. Ensuring that children develop appropriate relationships with PA, sedentary behavior, and sleep during this stage is crucial to establishing a foundation for lifelong health [21].

Although some studies have examined differences in ST and bedtime patterns between weekdays and weekends in preschoolers, they fail to account for variability within specific age groups or between sexes [22]. Our study seeks to bridge this gap by examining this developmental stage and assessing how early lifestyle behaviors shape future well-being [3,12].

To preserve and enhance the health of children under five years of age, the United Nations (UN) has issued general recommendations [23]. These include a minimum of 180 min of PA per day, with at least 60 min of moderate- to vigorous-intensity PA, 10–13 h of good-quality sleep, and less than 1 h of sedentary ST. These guidelines stand in stark contrast to the pervasive overuse of screens observed in both school and home environments. Additionally, the UN provides age-specific guidelines for healthy sleep durations: children aged 3–5 years should sleep 10–13 h (including naps), and children aged 6–13 years are advised to sleep 9–12 h nightly [23]. In terms of screen time, both the UN and the American Academy of Pediatrics (AAP) [24] recommend limiting ST for young children. For children aged 2–5 years, screen use should not exceed one hour on weekdays or three hours over the weekend. For children older than six years, ST should be limited to a maximum of 120 min daily, accompanied by the promotion of healthy habits and reduced engagement in screen-based activities.

Some researchers advocate for a more comprehensive approach to screen recommendations, emphasizing the need to account not only for the duration of ST but also for the nature of the content and the time of day at which it is consumed [15,25]. This perspective highlights the ongoing debate regarding the most appropriate criteria for assessing ST’s impact. Previous research has emphasized the importance of considering the context and content of screen exposure, suggesting that not all ST is equally detrimental [26,27]. The concept of “context and content” refers not only to the timing of screen exposure but also to the day of the week (weekdays or weekends) or individual characteristics like the child’s age and sex, which can influence the impact of screen exposure. It has been shown that excessive ST, especially before bedtime, disrupts sleep by affecting children’s melatonin production [18]. This highlights the importance of not only regulating ST duration but also focusing on the nature and timing of screen use, which has been underexplored in existing research on this age group.

While many studies have focused on excessive screen use among young children during the COVID-19 pandemic, these data face significant limitations, as children’s current lifestyles differ considerably from their experiences during the pandemic. Therefore, such findings may not accurately represent the present reality. Notably, these studies emphasize that parental decision-making largely determines screen-viewing practices at these ages. As such, educational initiatives regarding health recommendations from organizations concerned with children’s well-being should primarily target parents [21,22].

On the other hand, although numerous studies have examined the effects of ST, PA, and SL separately, few studies have comprehensively explored the interconnectedness of these three behaviors, particularly in young children [7]. The present study addresses a gap in the literature by examining the potential relationship between increased ST, reduced PA, and shorter SL in children aged 3–7 years. By adopting a holistic approach to analyzing this “behavioral triad”, our research offers a more nuanced understanding of how these behaviors interact and contribute to a cycle of sedentary habits. This integrated perspective represents a significant contribution to existing studies, which frequently analyze these factors in isolation. Moreover, despite the extensive body of research on this topic, a critical gap persists in studies focusing specifically on preschool and early elementary school children, particularly in analyses that account for both age and sex differences. Addressing these gaps will provide valuable insights for designing targeted strategies and interventions that promote balanced lifestyle behaviors in young children.

Finally, while much research on childhood lifestyles has been conducted in North America and Northern Europe, studies focusing on Spain remain limited. This study specifically examines children in a Mediterranean urban environment, providing valuable insights into how lifestyle behaviors are shaped within this district cultural, geographical, and socio-economic context. The urban setting provides unique conditions that may influence children’s behaviors, making the findings particularly relevant for understanding the impact of local factors on PA and sleep patterns. Gaining a deeper understanding of how to foster healthy lifestyle behaviors during this developmental stage is crucial for promoting long-term health [28].

Given the ongoing controversy and insufficient conclusive data, the aims of this study are (a) to examine the screen media use habits, sleep duration, and PA patterns of children aged 3 to 7 years, during the week and at the weekend, within an urban setting in Spain; (b) to identify how these behaviors differ between age groups (preschoolers vs. early elementary school children) and sex; and (c) to determine whether these behaviors align with the UN’s recommendations.

We hypothesize the following: (a) There are significant differences in ST, sleep duration, and PA patterns among children aged 3–7 years, according to age group. We expect that screen time will be higher in the older group, while sleep time and physical activity will be higher in the younger group. (b) There are significant differences in PA among children aged 3–7 according to sex, with boys being more active than girls, but no significant differences in ST and sleep duration according to sex. (c) Children’s weekday and weekend behaviors related to ST, SL, and PA will deviate from the recommendations set forth by the UN.

## 2. Materials and Methods

### 2.1. Study Design and Participants

A multicenter prospective cohort study was conducted with a sample of parents enrolled in the IISSAAR project “https://www.iissaar.com/about-us” (accessed on 7 November 2024) investigating the effects of digital media use on preschool children’s behaviors, including play, sleep, screen time, and eyesight, across urban areas in Asia and Europe.

Letters of invitation were distributed to various preschools and primary schools within the metropolitan area of Valencia, Spain. When a school expressed interest in participating, an information letter was provided to the parents or legal guardians. The inclusion criteria for participation were as follows: parents or legal guardians of children aged 3–7 years, parents who provided formal written consent, and those who completed the questionnaire in full. The exclusion criteria were (a) not living in the metropolitan area of Valencia, and (b) having children with medical conditions or functional diversity that could directly affect the data and introduce bias to the results.

A total of 243 parents fit our inclusion criteria and were analyzed in this study. The sample was divided into two groups based on the children’s ages, as reported by parents or legal guardians in the questionnaire: (a) Early Childhood Education, comprising children aged 3 to 5 years; and (b) Primary Education, comprising children aged 6 years and those who had recently turned 7 years but were born in the same calendar year as the 6-year-olds.

This study was conducted in accordance with the principles outlined in the Declaration of Helsinki, ensuring the protection of participants’ rights and well-being. The study was approved by the Research Ethics Committee of the Universidad Católica de Valencia (approval code: UCV/2028-2019/036) before data collection started. Since the study involved children, particular care was taken to meet ethical standards related to vulnerable populations. Parental consent was obtained before participation. Parents or legal guardians were provided with a detailed information sheet explaining the purpose of the study, the procedures involved, and the voluntary nature of participation. Written informed consent was secured to ensure that parents were fully aware of and agreed to their involvement in providing information on their children. Participants’ privacy was safeguarded throughout the study. Data collected using the Surveillance of Digital Media Habits in Early Childhood Questionnaire (SMALLQ^®^) were anonymized, ensuring that individual responses could not be traced back to specific participants. The data were stored securely and accessed only by authorized members of the research team.

### 2.2. Measurements

Data from parents were collected between October and December 2022 using the Surveillance of Digital Media Habits in Early Childhood Questionnaire (SMALLQ^®^) [1]. A forward and backward translation of the SMALLQ from English to Spanish was used according to the standardized World Health Organization-recommended protocol for cultural adaptation and language translation [29]. This validated tool consists of three sections and is designed to estimate children’s media habits—both on-screen and off-screen—as reported by parents for weekdays and weekends. The survey focuses specifically on behaviors occurring outside school hours.

For the present study, the variables extracted from the questionnaire included ST, SL, and PA during weekdays and weekends. These data were analyzed to identify differences based on age group and sex.

### 2.3. Data Synthesis and Analysis

A sensitivity analysis was performed using G*Power (version 3.1.) software [30] to determine the minimum effect size detectable in the study. For an alpha level (α) of 0.05 and a statistical power (1 − β) of 0.80 across four groups, the minimum detectable effect size was calculated as f = 0.180.

Statistical analyses were conducted using SPSS software (version 22) and JASP (version 0.18.3). A *p*-value of <0.05 was considered indicative of statistical significance. Descriptive statistics were calculated for all variables. To evaluate the effects of age group and sex on PA, ST, and SL during weekdays and weekends, a series of ANOVA tests were performed. Effect sizes (η^2^) were computed to quantify the proportion of variance explained by each factor.

## 3. Results

### 3.1. Participants’ Characteristics

A total of 243 parents were analyzed in this study. Figure 1 illustrates the flow diagram for the prospective cohort study.

### 3.2. Screen Time, Physical Activity, and Sleep Duration by Age and Sex

Descriptive data and ANOVA results are shown in Table 1.

ANOVA analyses of ST based on age group revealed a significant effect on weekdays (*F*(1, 241) = 4.348, *p* = 0.038, η^2^ = 0.018), with the preschool group spending more time on digital media, but no significant effect on weekends (*p* = 0.353). Gender did not have a significant impact on ST at either age (*p* > 0.05).

Regarding PA, age was found to significantly influence activity levels on both weekdays (*F*(1, 241) = 4.046, *p* = 0.045, η^2^ = 0.017) and weekends (*F*(1, 241) = 4.238, *p* = 0.041, η^2^ = 0.017), with younger children being more active. Gender had no significant effect on PA at any age (*p* > 0.05).

Finally, SL was significantly affected by age on weekends (*F*(1, 241) = 4.217, *p* = 0.041, η^2^ = 0.017), but not on weekdays (*p* = 0.579). Gender also had no significant effect on SL (*p* > 0.05).

### 3.3. Compliance with UN Recommendations

Based on the data provided, a comparison was made with the UN recommendations for ST, PA, and SL for children, categorized by age and sex (Table 2).

Among preschoolers aged 3 to 5, ST on weekdays significantly exceeds the recommended 60 min limit, averaging over 95 min for males and 107 min for females. On weekends, although the recommended limit is more lenient at 180 min, both groups approach this threshold, suggesting a persistent issue with sedentary behavior (males: 167 min; females: 163 min). For elementary school children aged 6 to 7, ST remains within the acceptable range of 120 min on weekdays, particularly for females, who average just 67.5 min. However, on weekends, both sexes exceed the 120 min recommendation, with males averaging 164 min and females 138 min, highlighting a pattern of increased media consumption on free days.

PA presents a contrasting trend. While preschoolers fall short of the recommended 180 min on weekdays, with males averaging 146 min and females 143 min, their activity levels improve considerably on weekends. Both groups surpassed the recommendation (males: 330 min; females: 297 min), indicating that weekends provide more opportunities for active play. However, this increase may not fully compensate for the weekday deficits. In terms of PA, elementary school children show mixed compliance. On weekdays, neither males nor females reach the recommended 180 min, averaging approximately 116 min. However, their activity levels improve significantly on weekends, with both groups engaging in well over the suggested amount (males: 248 min; females: 279 min).

SL patterns among preschoolers also raise concerns. On weekdays, both males and females sleep slightly below the minimum recommended 10 h, averaging around 9.7 and 9.6 h, respectively. Weekends show slight improvement, with average sleep durations slightly exceeding the lower limit of 10 h (males: 10.2 h; females: 10.1 h). SL among elementary school children remains consistently within the recommended range of 9 to 12 h per night. Both sexes maintain adequate sleep patterns on weekdays and weekends. On weekdays, males sleep 9.7 h and females 9.5 h. On weekends, the results show slight increases (males: 9.8 h; females: 9.9 h), suggesting that sleep routines may be more established at this age.

## 4. Discussion

This study examined children’s screen time habits, sleep patterns, and physical activity, considering the day of the week (weekday vs. weekend), age group (preschoolers vs. first grade of elementary school), and sex (males vs. females) in an urban Mediterranean context in Spain.

The findings align with existing literature, showing that age plays a more critical role than sex in shaping these lifestyle behaviors during early childhood. PA levels significantly decreased with age, which mirrors trends found in other studies [14,28,31]. This decline highlights the importance of promoting active play and sports in early childhood to cultivate healthy habits.

Age-related differences were evident across all the variables studied, with younger children engaging in more PA and sleeping longer on weekends compared to older children. Similarly, the decline in sleep duration with age observed in this study aligns with established developmental patterns, where sleep needs naturally decrease with age [6]. The reduced PA observed in the older group may be due to their structured daily routines, as elementary school children typically spend more time in organized activities compared to their younger counterparts. These structured schedules can limit opportunities for free play and active exploration, both of which are crucial for physical and cognitive development [21]. These results underline the necessity for interventions that promote PA and sports within this age group.

In contrast to previous studies reporting higher PA levels in boys than girls [26,32], no sex differences in PA or sleep were observed in this study. This absence of significant differences may be due to the relatively young age of the participants, as sex-based disparities in activity and sleep behaviors often become more pronounced in adolescence [33].

The study also supports previous findings that children’s screen time often exceeds the guidelines set by the American Academy of Pediatrics and the United Nations, contributing to adverse health effects such as reduced PA and shorter sleep durations [34,35]. Increased ST is linked to higher rates of sedentary behavior, which is consistently associated with lower physical fitness and a higher risk of obesity in children [4,5]. Research further indicates that prolonged screen exposure has detrimental effects, including not only reduced PA but also psychosocial and developmental challenges such as impaired social skills, lower academic performance, and behavioral problems [8,12]. However, this study acknowledges the potential benefits of screen media when used appropriately and in moderation, recognizing its role as a valuable educational resource that can support cognitive and communication development [2,27]. This highlights the need for more refined screen time guidelines [26] that consider both content type and context, rather than focusing solely on duration. Nonetheless, adherence to recommended ST limits is essential, especially as early childhood is a critical period for establishing healthy lifestyle habits.

According to the World Health Organization (WHO), children under the age of 5 should engage in at least 180 min of PA daily [23]. For preschool-aged children (3–5 years), the data largely comply with the UN recommendations for PA and sleep duration, confirming prior research [18]. However, ST exceeds the recommended limit on weekends, with both male and female preschoolers spending more than three hours per day on screens during this period, similar to results from previous studies [14].

Additionally, the findings suggest that a significant portion of children aged 5–6 do not meet the recommended sleep duration of 10–13 h. Moreover, for elementary school-aged children (6–13 years), the data indicate non-compliance with the PA recommendation, as children in this group engage in less than the required 180 min of daily activity, which aligns with previous research showing that younger children move more due to less structured routines [26]. While their ST remains within recommended limits, their sleep duration falls slightly below the 9–12 h guideline, yet still aligns closely with the lower end of the recommended range. This somewhat contrasts with previous studies [25,26] but reflects a more favorable outcome than expected. The decline in sleep duration with age also aligns with earlier findings, particularly on weekends [18]. These results support previous research indicating that sleep requirements decrease naturally as children grow older [36]. Older children tend to sleep less, both because they require less sleep and due to school schedules [18,25,26].

This study also found that younger children engaged in more PA and slept longer on weekends compared to older children. Insufficient sleep can contribute to various adverse outcomes, including behavioral issues, cognitive challenges, and an increased risk of obesity [9]. The relationship between excessive screen time and reduced sleep has been well documented in the literature [7]. This trend is consistent with previous studies [17,18], which have linked inadequate sleep to excessive screen time, particularly before bed.

Furthermore, this study emphasizes the interconnected nature of PA, ST, and SL, suggesting that excessive ST contributes to both reduced PA and disrupted sleep, creating a harmful cycle, as highlighted by other research [25]. Our findings suggest a negative loop, reinforcing sedentary behavior and contributing to the rising prevalence of childhood obesity and related health issues, particularly in preschoolers, while elementary school children show a more regulated balance. The study illustrates how these behaviors influence one another. For elementary school children, weekday ST generally meets recommendations, but PA remains insufficient, likely due to structured school routines. On weekends, increased ST is associated with higher PA levels, showing a more balanced pattern compared to preschoolers. Notably, sleep duration in this group remains consistent and within recommended ranges, suggesting that older children’s sleep routines are less affected by fluctuations in ST and PA. These findings underline the need for interventions to reduce screen exposure, particularly during the hours leading up to bedtime, to promote healthier sleep habits, as demonstrated by earlier studies [6,25]. Excessive screen use before bedtime can disrupt melatonin production, which is essential for regulating sleep cycles [4]. This disruption may result in delayed sleep onset, reduced sleep duration, or poorer sleep quality, which may explain why children with higher ST tend to experience less sleep.

Excessive ST not only reduces opportunities for PA and sports but also disrupts sleep patterns, fostering a sedentary lifestyle and weight gain. This interdependence of behaviors has been corroborated by other studies, which highlight the significant short- and long-term impacts on children’s well-being [6].

Given the relationship between these variables, a comprehensive and multidimensional approach is crucial to promote healthier lifestyle behaviors in young children. Interventions should focus on increasing opportunities for physical activity, ensuring sufficient sleep, and addressing excessive screen time. Engaging parents and legal guardians is essential, as they play a key role in establishing and reinforcing healthy routines. These habits should prioritize regular PA, limited ST, and the creation of an optimal sleep environment. To break the cycle of sedentary behavior and associated health risks, interventions must target both children and their caregivers. Strategies should include setting clear limits on ST, encouraging active play, and promoting consistent and adequate sleep patterns. By fostering an environment that prioritizes health and well-being, these efforts can significantly improve children’s long-term lifestyle behaviors and overall quality of life.

This study achieved all its objectives. The findings confirm the primary hypotheses that significant age-related differences exist in PA, SL, and ST among children aged 3–7 years, and that these behaviors often deviate from recommended health guidelines. The hypothesis that younger children would engage in more PA and sleep longer compared to older children was supported, as was the expectation that excessive ST would correlate with reduced PA and shorter SL. The results partially support hypothesis (a), showing that age influences ST, PA, and SL. The second hypothesis (b), which proposed significant sex-based differences in PA, was not supported, as no significant variations were observed. The third hypothesis (c) is mostly confirmed, as children’s ST exceeded recommendations, PA was insufficient on weekdays, and SL was slightly below guidelines for preschoolers.

This study has several methodological limitations that should be considered when interpreting its findings. First, the sample is geographically limited to an urban Mediterranean context (Valencia, Spain). The geographical, climatic, and socio-economic characteristics of the analyzed context may influence the generalizability of the results, as these variables act as moderators of the dependent variables studied. Furthermore, the data on dependent variables (ST, PA, and SL) are based on parental reports, which may be subject to recall bias or inaccuracies, potentially affecting the reliability of the data. Another limitation of the study is its cross-sectional design, which captures a single snapshot of behaviors at a specific point in time. This limits the ability to draw conclusions about causal relationships. Additionally, the focus on children aged 3 to 7 years restricts the findings to a specific developmental stage, which may not apply to children in other age groups.

Given these limitations, future research should address several key areas. Expanding the sample to include a larger, more diverse group of participants would enhance the generalizability of the findings. Using objective measures, such as device usage tracking or direct observation, would provide more accurate data on screen time, physical activity, and sleep patterns. A longitudinal approach would be valuable for examining long-term effects and causal relationships.

Future studies should also explore the long-term impact of screen time on both physical and mental health. Investigating the effects of different types of screen content—educational versus recreational—on development could provide important insights. Finally, developing comprehensive guidelines that consider both the duration and purpose of screen use could help mitigate the potential negative impacts of screen time on young children’s well-being.

## 5. Conclusions

This study shows that age, rather than sex, significantly influences physical activity, screen time, and sleep habits in preschool and first-grade children. Younger children engage in more physical activity and have longer sleep durations on weekends compared to older children, while excessive screen time is linked to lower sleep time and reduced physical activity. The study also found that children’s ST often exceeded recommendations from the American Academy of Pediatrics and the UN, particularly on weekends.

## Figures and Tables

**Figure 1 sports-13-00091-f001:**
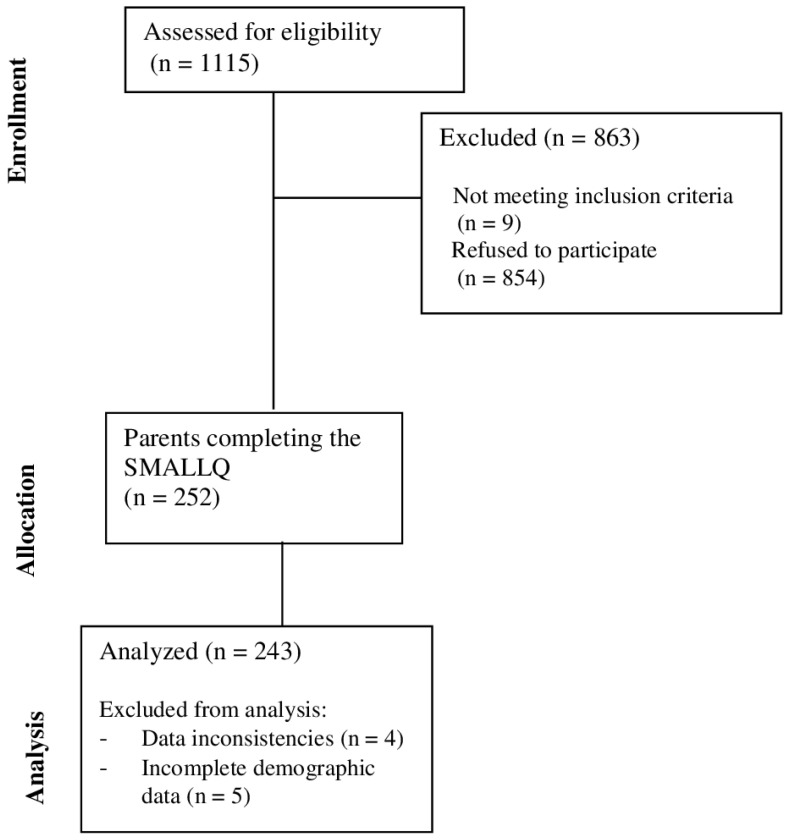
Flow diagram for the prospective cohort study.

**Table 1 sports-13-00091-t001:** Sex and age group differences in screen time, physical activity, and sleep time.

		ST (min)		PA (min)		SL (min)	
**Sex**	Age	Weekday *	Weekend	Weekday *	Weekend *	Weekday	Weekend *
**Male** **(n = 113)**	Preschoolers (n = 65)	95.3 (±93.0)	166.6 (±158.0)	146.1 (±121.4)	330.0 (±171.4)	580.1 (±85.2)	613.2 (±90.5)
	Elementary (n = 48)	85.6 (±72.2)	164.1 (±108.6)	116.3 (±97.8)	247.5 (±156.5)	581.7 (±51.5)	589.3 (±40.8)
**Female (n = 129)**	Preschoolers (n = 68)	107.2 (±136.7)	163.1 (±132.5)	142.7 (±104.2)	296.9 (±205.8)	578.5 (±52.0)	603.3 (±59.7)
	Elementary (n = 61)	67.5 (±56.4)	138.6 (±89.9)	116.2 (±118.5)	279.2 (±188.8)	569.6 (±53.2)	592.8 (±60.4)

Note: ST (screen time), PA (physical activity), SL (sleep time); *: statistically significant differences *p* < 0.05 between age groups.

**Table 2 sports-13-00091-t002:** Comparison with UN recommendations.

Age Group	UN Recommendations	Sex	Day	ST	PA	SL
Preschoolers (3–5 years)	ST: 60 min/day (weekdays), 180 min/day (weekends) PA: 180 min/day SL: 10–13 h/day	Male	Weekdays	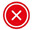	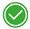	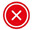
Weekends	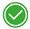	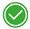	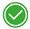
Female	Weekdays	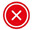	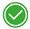	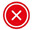
Weekends	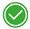	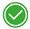	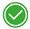
Elementary (6–7 years)	ST: 120 min/day PA: 180 min/day SL: 9–12 h/day	Male	Weekdays	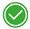	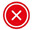	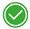
Weekends	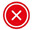	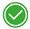	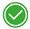
Female	Weekdays	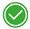	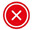	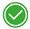
Weekends	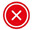	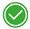	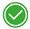

Note: ST (screen time), PA (physical activity), SL (sleep time); 
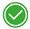
 (accomplished); 
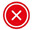
 (Not accomplished).

## Data Availability

https://osf.io/ku62y/?view_only=74ed97e3fe8e46308ef36a1358039e69 (accessed on 7 March 2025).

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
