# Peer review of "Associations Between Screen Time, Physical Activity, and Sleep Patterns in Children Aged 3–7 Years—A Multicentric Cohort Study in Urban Environment"

_sports, 2025, doi:10.3390/sports13040091_

Round 1
Reviewer 1 Report
Comments and Suggestions for Authors
It is a valuable and well-designed study on an important topic. It provides useful information about how screen time, physical activity, and sleep patterns are connected in young children. However, there are some areas where the article could be improved to make it even better:
1) the introduction should include the study’s hypotheses and research questions. Adding these would make the purpose of the study clearer and help readers understand what the authors aimed to investigate. The research questions would also provide a stronger structure for the study.
2) the discussion section should explain whether the hypotheses were confirmed and if the research questions were answered. This would connect the results back to the study’s goals and make the conclusions easier to follow.
3) the authors should add information about how the study followed ethical guidelines, especially those described in the Publication Manual of the American Psychological Association. Since the study involves children, it is important to explain how parental consent was obtained, how participants’ privacy was protected, and how ethical standards were met.
4) the article would benefit from more discussion about how the findings compare to other studies on similar topics. This would help readers understand how the results fit into the bigger picture and why they are important.
I’m keeping my fingers crossed for good revisions and the final publication of the article.
Author Response
Attached is a file with the responses to reviewer 1.

Reviewer 2 Report
Comments and Suggestions for Authors
It is a pleasure to review your manuscript, "Associations Between Screen Time, Physical Activity, and Sleep Patterns in Children Aged 3-7 Years: A Multicentric Cohort Study in Urban Environment." The study explores a relevant and timely topic with significant academic and practical value. Your efforts are commendable, and the manuscript shows promise. However, some revisions and clarifications are required to enhance its quality and ensure its contribution to the field is maximized.
1. While the introduction highlights the broader issues of screen time, physical activity, and sleep patterns, it fails to succinctly define the specific gaps this study addresses within the 3–7-year age group. A more focused presentation of how this study distinguishes itself from existing literature is necessary to establish its contribution effectively.
2. The manuscript lacks a dedicated literature review section, which is essential for systematically contextualizing the study within existing research and identifying specific gaps it aims to address.
3. The sample size of 243 parents, while sufficient for some statistical analyses, limits the generalizability of the findings to broader populations, particularly across diverse urban and rural settings. Future studies should incorporate a larger and more demographically diverse sample.
4. While ANOVA results are presented for age and sex differences, the analysis does not delve into potentially meaningful interactions between screen time, physical activity, and sleep patterns.
5. The discussion overly reiterates established findings about the detrimental effects of excessive screen time, physical inactivity, and insufficient sleep, offering little novel interpretation into the observed relationships.
Author Response
Attached is a file with the responses to reviewer 2.

Reviewer 3 Report
Comments and Suggestions for Authors
The study is very interesting and important. I found mostly minor flaws:
1. The Authors have to define clear aim of the study at the end of Introduction.
2. Please report your study strictly in accordance with the STROBE Statement. Please use the STROBE checklist for your type of observational study.
3. The Authors have to report the exclusion criteria for study participants within Materials and Methods.
4. The Authors should present the latest studies (in Introduction or Discussion) related to sleep pattern and sleep habits among children which are very important for pediatric sleep medicine: doi:10.17219/dmp/167411, doi:10.17219/dmp/150615
Author Response
Attached is a file with the responses to reviewer 3.
